# Domain Generalization for Robust Model-Based Offline Reinforcement Learning

**Alan Clark, Shoaib Ahmed Siddiqui, Usman Anwar, Stephen Chung, David Krueger**
Department of Engineering, University of Cambridge, Cambridge, CB2 1PZ, UK
{ajc348,msas3,ua237,mhc48,dsk30}@cam.ac.uk

**Robert Kirk**
Centre for Artificial Intelligence, University College London, London, WC1V 6LJ, UK
robert.kirk.20@ucl.ac.uk

## Abstract

Existing offline reinforcement learning (RL) algorithms typically assume that training data is either: 1) generated by a known policy, or 2) of entirely unknown origin. We consider multi-demonstrator offline RL, a middle ground where we know which demonstrators generated each dataset, but make no assumptions about the underlying policies of the demonstrators. This is the most natural setting when collecting data from multiple human operators, yet remains unexplored. Since different demonstrators induce different data distributions, we show that this can be naturally framed as a domain generalization problem, with each demonstrator corresponding to a different domain. Specifically, we propose Domain-Invariant Model-based Offline RL (DIMORL), where we apply Risk Extrapolation (REx) [15] to the process of learning dynamics and rewards models. Our results show that models trained with REx exhibit improved domain generalization performance when compared with the natural baseline of pooling all demonstrators' data. We observe that the resulting models frequently enable the learning of superior policies in the offline model-based RL setting, can improve the stability of the policy learning process, and potentially enable increased exploration.

## 1 Introduction

In the standard online reinforcement learning (RL) paradigm, agents often require millions of interactions with the environment in order to learn an optimal policy for completing a task. In many settings, however, the process of exploration in the real-world is undesirable, impractical or unsafe [17, 19]. It would therefore be preferable to enable learning from *demonstrations*, rather than direct *interaction* with the environment. Datasets targeting activity recognition from ego-centric videos [8] highlight the prevalence and ease of collecting demonstrations in our setting. Demonstrations are provided by one or more *demonstrators*, each executing their own behavioural policy, $\pi_B$–their method of achieving the task. Various approaches to performing offline RL have been proposed [17, 19], however, to the best of our knowledge, none of these have exploited information regarding the collection of data from multiple demonstrators.

A fundamental challenge in off-policy RL is policy-induced *distributional shift*; this is especially problematic in offline RL [17, 28]. Standard off-policy methods often fail when trained using offline data [10]. Value-based offline RL algorithms have addressed this by encouraging the learning agent to stay close to the behaviour policy that generated the training data. Such a constraint can (provably) limit performance, however Buckman et al. [4], and other model-based approaches, which do not suffer from this weakness, have been found to outperform value-based methods [28, 14].

3rd Offline Reinforcement Learning Workshop at Neural Information Processing Systems, 2022.

Rather than constraining the learning agent's exploration, the aim of our work is to increase the robustness of *environment models*[1] to distributional shifts by applying *Risk Extrapolation (REx)* [15] during their training. Our hope is that this will enable learning policies to explore more of the state-action space without incurring significant training instability. REx assumes that training data from multiple domains (in our case, demonstrators) is available. While the demonstrator that generated each training record is known, no knowledge about the demonstrator's policy is required. Even if distributional shifts more extreme than those observed at training time are encountered, REx aims to achieve similar risks on out-of-distribution domains by encouraging the training losses/risks across the domains present in the training data to be equal [15].

The primary contribution of our work is a practical algorithm for performing **Domain-Invariant Model-based Offline RL (DIMORL)** using multi-demonstrator datasets. We empirically show that environment models trained with REx exhibit improved average and worst-case performance on out-of-distribution datasets. Furthermore, in a majority of cases, policies trained offline using these models attain higher average returns. We demonstrate that environment models trained with REx can enable more stable offline model-based policy learning, and potentially support increased exploration. We present hypotheses for the benefits observed, and share the supporting evidence gathered thus far. Further experimentation and analysis are needed to draw firm conclusions.

## 2 Background and Related Work

**Offline Reinforcement Learning** In offline RL, an agent $\pi$ is trained using *only* a fixed dataset $\mathcal{D}$ of transitions ($\{s, a, r, s'\}$) [10], or trajectories ($\{s_1, a_1, r_1, ..., s_T, a_T, r_T\}$) [1], without any further interaction with the environment [17, 19]. The data may also include meta-data about the policies used to collect the data. Our aim is to learn the optimal policy, $\pi^*(a|s)$, that maximises the expected sum of discounted rewards $\pi^* = \arg\max_\pi \mathbb{E}_\pi[\sum_{t=0}^\infty \gamma^t r(s_t, a_t)]$, using discount factor $\gamma \in (0, 1]$.

**Distributional Shift** While the transition distribution of a Markov-Decision Process (MDP) is independent of the policy being followed, the state and state-action visitation distributions induced by behavioural policy $\pi_B$, $d^{\pi_B}(s)$ and $d^{\pi_B}(s, a)$ respectively, are not. The data collected for offline policy training is typically assumed to be composed of *iid* samples drawn from $d^{\pi_B}(s, a)$, and is likely to cover only a fraction of the total state-action space. In order to learn optimal policies, we would like our learning algorithm to *generalize* to other areas of this space; departing from the support of the training data in order to learn good behaviours that are not exhibited in the static dataset [28, 27]. The policy being learned, $\pi_{\text{off}}$, would therefore induce new state and state-action visitation distributions, $d^{\pi_{\text{off}}}(s)$ and $d^{\pi_{\text{off}}}(s, a)$. Errors made by the learned environment models, such as those arising from poor generalization performance under distributional shifts, can result in *model exploitation*: the policy being trained learns to take advantage of model errors when optimizing the reward, leading to poor performance in the real environment [5, 20, 13, 7, 17]. However, in more extreme cases, model exploitation can cause significant instability in training, preventing a policy from being learned [16, 18].

**Offline Model-Based Reinforcement Learning** Kidambi et al. [14] and Yu et al. [28] proposed the model-based offline RL algorithms MoREL and MOPO respectively. Both of these methods are conservative in the face of model uncertainty. However, they do not explicitly penalize deviations from the demonstrator's distribution, thus avoiding performance limitations demonstrated by Buckman et al. [4]. Both MoREL and MOPO work by modifying a learned MDP to penalize uncertain transitions. MoREL's pessimistic MDP adjusts $P(s'|s, a)$ to transition to a low-reward terminal state when the dynamics of $(s, a)$ are uncertain, as measured by disagreement within an ensemble of learned dynamics models [14]. In MOPO, Yu et al. [28] instead penalize uncertain transitions directly in the reward function, and measure uncertainty as the maximum standard deviation across members of an ensemble. MOPO extends MBPO [13], which utilizes learned environment models to generate short roll-outs of length $h$ that are employed to update the learning policy using the soft-actor critic (SAC) policy gradient algorithm [11, 12].

---

[1]We use the term *environment model* to highlight that both dynamics and rewards models are learned. We do not learn an initial state distribution. During policy training, transitions are generated from starting locations sampled from an offline dataset. When evaluating policies, the real initial state distribution is used.

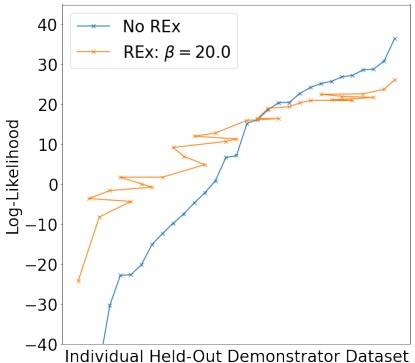 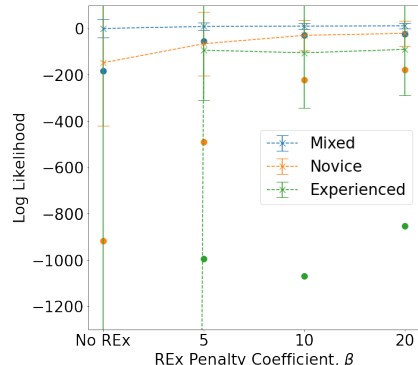

Figure 1: Environment models trained against HalfCheetah multi-demonstrator datasets were used to calculate the log-likelihoods of a selection of evaluation datasets, each generated by an individual held-out demonstrator. **Left:** The environment model trained with REx ($\beta = 20$) exhibits more consistent performance across the individual held-out demonstrator datasets. Models were trained against the *Mixed* multi-demonstrator dataset. **Right:** REx increases the average (crosses and dashed lines; error bars indicate one standard deviation) and worst-case (dots) log-likelihoods. Environment models were trained against *Novice*, *Mixed* and *Experienced* multi-demonstrator datasets.

## 3 Methods

Our method stems from the observation that different demonstrators in an offline RL setting correspond to different domains in a domain generalization setting. We propose the **Domain-Invariant Model-based Offline RL (DIMORL)** algorithm, which supplements model-based offline RL techniques by applying Risk Extrapolation (REx) [15] when training environment models.

### 3.1 Risk Extrapolation (REx)

The Risk Extrapolation (REx) domain generalization method aims to enforce strict equality of risks (i.e., losses) across training domains [15]. This helps guarantee that a model trained with REx will be invariant; i.e., it should also achieve (roughly) the same loss on test domains that are in the *affine span* of the training domains. This is demonstrated in Fig. 1, which shows that an environment model trained with REx achieves more consistent performance when evaluating log-likelihoods across a range of datasets that were each generated by a different held-out demonstrator (i.e., demonstrators that did not contribute to the model's training data). We use the simple Variance-REx (V-REx) algorithm, which penalizes the variance of the training risks:

$$\mathcal{R}_{\text{V-REx}}(\theta) \doteq \beta \, \mathrm{Var}(\{\mathcal{R}_1(\theta), ..., \mathcal{R}_M(\theta)\}) + \sum_{e=1}^{M} \mathcal{R}_e(\theta) \tag{1}$$

where $\mathcal{R}_e$ is the risk on the $e$-th domain, and $\beta$ controls the strength of the variance regularizer. The robustness of REx to multiple forms of distributional shift [15] make it a strong candidate for our investigations, however other domain generalization techniques could be explored [2, 21, 26, 23, 30].

### 3.2 Domain-Invariant Model-based Offline RL (DIMORL) Implementation

In this work, we implement a version of DIMORL that extends the MOPO algorithm [28] by applying REx during environment model training. Experiments are performed using $\beta \in \{0, 5, 10, 20\}$, $\lambda \in \{0, 1, 5\}$ and $h \in \{5, 10\}$, except where otherwise stated. When $\beta = 0$ the MOPO algorithm is recovered, and when additionally $\lambda = 0$ the MBPO algorithm is recovered, providing two baselines against which our results can be compared. Ensembles of environment models, $p_{\theta,\phi}(s_{t+1}, r_t | s_t, a_t)$, are trained against multi-demonstrator datasets, $\mathcal{D}_{\mathcal{E}} = \bigcup_{e=1}^{M} \mathcal{D}_e$, where the outputs of model $i$ parameterize a Gaussian distribution with diagonal covariance matrix: $\mathcal{N}(s_{t+1}, r_t; \mu_\theta^i(s, a), \Sigma_\phi^i(s, a))$. Individual risks, $\mathcal{R}_e$, are calculated for each demonstrator's data, $\mathcal{D}_e$, and REx used to reduce the variance of the risks. Thus, we take advantage of knowing which demonstrator generated each record, but assume no knowledge of any demonstrator's policy. See Appendix A for further details.

Table 1: Environment models trained with REx (i.e., where $\beta > 0$) yielded policies with the highest average returns in two-thirds of datasets across three MuJoCo environments. For the *Novice* multi-demonstrator datasets and Hopper *Mixed* dataset, the policies learned using DIMORL had higher average returns (bolded values) than any of the individual demonstrators used to generate the dataset. The average policy evaluation returns ± one standard deviation over three random seeds are shown, along with the REx penalty coefficient $\beta$, MOPO penalty coefficient $\lambda$, and roll-out length $h$ used.

| | Mult-Demonstrator Dataset | | | | | | | | |
| | Novice | | | Mixed | | | Experienced | | |
| Environment | Max Dem. Return | DIMORL Return | $(\beta, \lambda, h)$ | Max Dem. Return | DIMORL Return | $(\beta, \lambda, h)$ | Max Dem. Return | DIMORL Return | $(\beta, \lambda, h)$ |
|---|---|---|---|---|---|---|---|---|---|
| HalfCheetah | 7663 | **9056** ± 122 | (0,1,5) | 11032 | 6687 ± 4888 | (0,1,5) | 14511 | 4547 ± 288 | (10,5,10) |
| Hopper | 3239 | **3482** ± 23 | (20,5,5) | 3553 | **3569** ± 40 | (20,5,50) | 3553 | 3493 ± 33 | (0,5,50) |
| Walker2D | 1587 | **1594** ± 1598 | (20,0,10) | 5430 | 2265 ± 141 (2) | (20,0,5) | 5430 | 4796 ± 113 | (20,0,10) |

# 4 Experiments

As a proxy for real-world demonstrators, we trained a selection of policies online against the OpenAI Gym MuJoCo Hopper, HalfCheetah and Walker2d environments [3, 25] using the soft actor-critic (SAC) algorithm [11, 12]. Subsets comprising five of these pseudo-demonstrators were used to generate three multi-demonstrator datasets for each environment: *Novice*, *Mixed*, and *Expert*. See Appendix B for details of the SAC pseudo-demonstrators and multi-demonstrator datasets produced.

## 4.1 Domain Generalization Performance of Environment Models

We found that environment models trained with REx achieved higher average and worst-case out-of-distribution (OOD) log-likelihoods for each environment and multi-demonstrator dataset. Across all environments, OOD performance was worst for the *Experienced* datasets, and was most greatly improved by the incorporation of REx to model training. This can be seen for the HalfCheetah environment in Figure 1. The results for the other environments can be found in Appendix C. The average and worst case performance generally continued to improve as the REx penalty coefficient was increased from 5 to 20. While this might encourage the use of larger coefficients, we did not find the domain generalization performance of the models to be well correlated with the average returns of policies trained using them. This is discussed further in Appendix C.1.

## 4.2 Offline Agent Training

Policies learned using environment models trained with REx obtained the highest average return for two-thirds of multi-demonstrator datasets across the MuJoCo environments investigated. Of these, the largest REx penalty coefficient investigated ($\beta = 20$) yielded the highest return in all but one setting. Table 1 provides an overview of the hyperparameter settings used to achieve the highest performing policies. The full results are discussed in Appendix D.1.

We additionally found that environment models trained with REx were more robust to noisy initial state distributions; highlighting the benefits that improved domain generalization performance can bring to policy training (see Appendix F.1). We further attempted to learn policies using roll-out starting locations sampled randomly from datasets not used to train the environment models, with the expectation that environment models trained with REx would exhibit improved robustness to such distributional shifts. However, no good policies were learned under any settings investigated (see Appending D.2), likely due to the distributional shifts encountered being simply too extreme.

## 4.3 Increased Policy Training Stability

In an attempt to pinpoint the sources of the benefits brought by environment models trained with REx, we analyzed the agent training performance and found that, for the HalfCheetah environment, the Q-functions of soft actor-critic policies were often less prone to becoming degenerate when using REx penalty coefficients $\beta \in \{10, 20\}$. This is demonstrated in Fig. 2. We present two hypotheses regarding the source of the Q-function instability, and why REx may be yielding improvements.

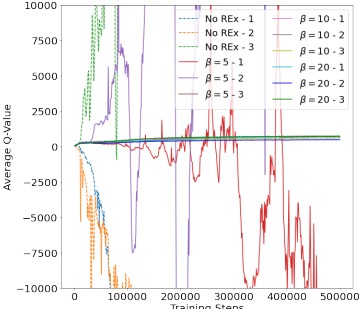 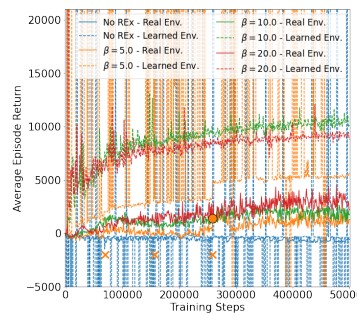

Figure 2: Average Q-Values (**left**) and episode returns (**right**) during policy training for the *Mixed* HalfCheetah multi-demonstrator dataset, with $\lambda = 0$ and $h = 10$. **Left:** Q-functions learned during SAC policy training were less likely to become degenerate when using environment models trained with larger REx penalty coefficients ($\beta \in \{10, 20\}$). For each value of $\beta$, individual results across three random seeds are shown. **Right:** When evaluating policies against the learned environment model used to train them (dashed lines), extreme episode returns were observed for models trained using no or lower penalties ($\beta \in \{0, 5\}$). Average values across three random seeds are shown.

**Degenerate Predictions:**  The magnitude of the evaluation returns in Figure 2 highlight that the learned reward models can make degenerate predictions. This could lead to instability in Q-function training. However, the returns shown were obtained over 1000 step episodes, whereas roll-outs of $h \in \{5, 10\}$ steps were typically generated for policy training. The reward models would therefore need to make degenerate predictions within 5-10 steps to negatively impact Q-function learning. We have observed cases of this (see Appendix E), however it does not appear to be a necessary condition for Q-function instability to occur. Note, rather than the issue lying solely with the reward models, it is likely that errors in the dynamics models will also contribute by leading the agent into unnatural states. Further analysis of the trajectories produced in the learned environment is necessary.

**Environment Model Complexity:**  The learned environment model may be less well-behaved and more complex than the real environment. For instance, a small change in the current state could have a large and unpredictable impact on the predicted next state. If we assume the neural network used has sufficient capacity and is provided with sufficient data to capture the real environment, the function approximation component of the classic deadly triad [24] would theoretically not be an issue. However, the environment model state representation may fail to capture all quirks in the model, resulting in function approximation continuing to be problematic.

### 4.4   Increased Exploration During Policy Training

We hypothesize that the improved domain generalization performance of environment models trained with REx may enable increased exploration of the state-action space during policy training, yielding more optimal policies. Our hope is that policy training will continue to remain stable, and model exploitation will be minimized. To visualize the extent of exploration, we use PCA to project transition records into two-dimensions. Initial experiments potentially indicate increased exploration (see Appendix F.2), however further work is necessary to ensure our observations cannot simply be explained by the fraction of the original data's variance that is captured by the projection.

## 5   Conclusions

We have highlighted and exploited the opportunity to take advantage of knowledge regarding which demonstrators generated offline datasets, without assuming any information about each demonstrator's policy. We presented a practical algorithm for Domain-Invariant Model-based Offline RL (DI-MORL), and have illustrated that this method may improve the stability of policy learning, enable increased exploration, and yield more optimal policies in many settings. Further experimentation and analysis is needed to validate our findings, and to fully take advantage of the algorithm's potential. The flexibility of our approach enables the combination of many alternative domain generalization techniques and model-based RL algorithms, presenting exciting avenues for future investigation.

## Acknowledgments

We thank Amy Zhang and Scott Fujimoto for their efforts and involvement in an earlier version of this project.

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

# Appendices

## A  DIMORL Implementation Details

The MOPO algorithm [28] and codebase[2] is used as the basis for our practical implementation of DIMORL. Our modified version is outlined in Algorithm 1. We train ensembles of environment models, $p_{\theta,\phi}(s_{t+1}, r_t | s_t, a_t)$, against multi-demonstrator datasets $\mathcal{D}_{\mathcal{E}} = \bigcup_{e=1}^{M} \mathcal{D}_e$, where the outputs of the models parameterise a Gaussian distribution with diagonal covariance matrix: $\mathcal{N}(s_{t+1}, r_t; \mu_\theta^i(s, a), \Sigma_\phi^i(s, a))$.

Individual negative log-likelihood values, $\mathcal{R}_{nll}^e$, are calculated for each demonstrator's data, $\mathcal{D}_e$. These are then used to calculate the MOPO V-REx loss given by Equation 2, where $\beta$ is the REx penalty coefficient. $\mathcal{R}_{wr}$ is an $\ell_2$ weight regularisation penalty, and $\mathcal{R}_{vb}$ is a loss term relating to the learning of variance bounds. Chua et al. [6] highlight that outside of the training distribution the predicted variance can assume arbitrary values, and can both collapse to zero or explode to infinity (in contrast to models like GPs where variance values are better behaved). Chua et al. [6] found that bounding the output variance such that it cannot exceed the range seen in the training data was beneficial, and so include the $\mathcal{R}_{vb}$ penalty. The weight regularisation and variance bounding terms also appear in the loss function used by MOPO to train environment models [28].

$$\mathcal{R}_{MOPO\ V\text{-}REx} = \sum_{e=1}^{M} [\mathcal{R}_{nll}^e] + \beta \cdot \text{Var}\left(\mathcal{R}_{nll}^1, \mathcal{R}_{nll}^2, \ldots, \mathcal{R}_{nll}^M\right) + \mathcal{R}_{wr} + \mathcal{R}_{vb} \tag{2}$$

---

**Algorithm 1** DIMORL - an extension of MOPO[28] to include REx in environment model training

---

**Require:** : MOPO reward penalty coefficient $\lambda$, roll-out length $h$, roll-out batch size $b$, **REx penalty coefficient** $\beta$

1: Train on batch data $\mathcal{D}_{\mathcal{E}}$ an ensemble of $N$ probabilistic dynamics models $\{p_{\theta,\phi}(s_{t+1}, r_t | s_t, a_t) = \mathcal{N}(\mu_\theta^i(s, a), \Sigma_\phi^i(s, a))\}_{i=1}^N$ **with REx penalty coefficient** $\beta$
2: Initialise policy $\pi^{off}$ and empty replay buffer $\mathcal{D}_{\text{model}} \leftarrow \varnothing$
3: **for** epoch $1, 2, \ldots$ **do**
4:    **for** $1, 2, \ldots, b$ (in parallel) **do**
5:        Sample state $s_1$ from $\mathcal{D}_{\mathcal{E}}$ for the initialisation of the roll-out
6:        **for** $j = 1, 2, \ldots, h$ **do**
7:            Sample an action $a_j \sim \pi(s_j)$
8:            Randomly pick dynamics $\hat{T}$ from $\{\hat{T}^i\}_{i=1}^N$ and sample $s_{j+1}, r_j \sim \hat{T}(s_j, a_j)$
9:            Compute $\tilde{r}_j = r_j - \lambda \max_{i=1}^N \|\Sigma_\phi^i(s_j, a_j)\|_F$
10:           Add sample $(s_j, a_j, \tilde{r}_j, s_{j+1})$ to $\mathcal{D}_{\text{model}}$
11:   Drawing samples from $\mathcal{D}_{\mathcal{E}} \bigcup \mathcal{D}_{\text{model}}$, use SAC to update $\pi^{off}$

---

Environment model training was split into two phases:

1. **ERM:** No REx penalty is used (i.e., $\beta = 0$) until the original MOPO termination condition is reached: the loss on a held-out evaluation dataset does not decrease by more than 1 % for any model in the ensemble for five consecutive epochs [28].

2. **REx:** Training was then allowed to continue for the same number of epochs completed in the ERM phase, with the REx penalty coefficient now set to a user defined value. We investigated REx penalty coefficients $\beta \in \{0, 5, 10, 20\}$ in our work.

## B  SAC Demonstrators and Multi-Demonstrator Datasets

To proxy for real-world demonstrators (whether humans, or pre-existing control policies), we train a selection of demonstrator policies, $\pi^e$, using online RL algorithms and the OpenAI Gym MuJoCo

---

[2]`https://github.com/tianheyu927/mopo`

Table 2: Two implementations (Softlearning [11, 12] and D3RLPY [22]) of the online soft actor-critic (SAC) algorithm [11, 12] are used to create pseudo-demonstrators by taking snapshots at regular intervals during training to emulate demonstrators (with distinct policies) of varying degrees of skill. As would be expected, the return of the policies generally increases with the amount of training.

| Environment | Online Training Steps (millions) | Policy Return | |
|---|---|---|---|
| | | Softlearning | D3RLPY |
| HalfCheetah | 0.1 | 5126 | 3744 |
| HalfCheetah | 0.2 | - | 5050 |
| HalfCheetah | 0.25 | 7663 | - |
| HalfCheetah | 0.5 | 9324 | 7225 |
| HalfCheetah | 1 | 11032 | 8300 |
| HalfCheetah | 2 | 14511 | 9022 |
| HalfCheetah | 3 | 14215 | - |
| Hopper | 0.1 | 667 | 2567 |
| Hopper | 0.2 | 3239 | 1974 |
| Hopper | 0.4 | 3414 | 2395 |
| Hopper | 0.6 | 3051 | 2158 |
| Hopper | 0.8 | 3524 | 2625 |
| Hopper | 1.0 | 3553 | 3179 |
| Walker2d | 0.1 | 319 | 369 |
| Walker2d | 0.2 | - | 1587 |
| Walker2d | 0.25 | 1225 | - |
| Walker2d | 0.5 | 3004 | 3900 |
| Walker2d | 1 | 4139 | 4312 |
| Walker2d | 2 | 4134 | 5430 |
| Walker2d | 3 | 4795 | - |

Hopper, HalfCheetah and Walker2d environments. The sole purpose of these policies is to generate individual demonstrator datasets, $\mathcal{D}_e$, for use in training environment models. Each dataset comprises $N_e$ transition tuples of the form: $(s^e, a^e, s'^e, r^e, e)$. The unique demonstrator identifier $e$ is used to identify individual domains during the training of dynamics models with Risk Extrapolation (REx). The identifier is simply a number, and provides no information about the policy followed by the demonstrator. Independently sampled evaluation datasets are also created for each demonstrator. Collections of $M$ individual datasets are combined to produce multi-demonstrator datasets: $\mathcal{D}_\mathcal{E} = \{(s_i^e, a_i^e, s_i'^e, r_i^e, e)_{i=1}^{N_e}\}_{e=1}^M$.

A selection of soft actor-critic (SAC) [11, 12] policies are trained online. To increase the diversity of the demonstrators, we use two different implementations of the SAC algorithm: **Softlearning**, the official SAC implementation [11, 12]; and **D3RLPY**, which implements a version of SAC with delayed policy updates [22]. Policies are trained using the default hyperparameters provided in their respective repositories. As shown in Table 2, snapshots are taken at regular intervals during training to emulate demonstrators (with distinct policies) of varying degrees of skill. Table 3 shows the individual pseudo-demonstrator policies obtained. To further increase data diversity, additional demonstrator datasets are created using a random policy, $\pi^\mathcal{N}$.

We select individual subsets of five pseudo-demonstrators (i.e., five distinct policies) to produce three multi-demonstrator datasets for each MuJoCo environment: *Novice*, *Mixed*, and *Experienced*. As their names indicate, the *Novice* and *Experienced* datasets comprise transition records generated using pseudo-demonstrators trained online for minimal and extended number of training steps respectively, while *Mixed* uses a combination of each. Each pseudo-demonstrator contributes 20,000 transition records, and so each multi-demonstrator dataset consists of a total of 100,000 records.

In future work, we would look to train pseudo-demonstrators using a greater variety of online RL algorithms to further increase the diversity of policies used.

Table 3: Individual demonstrator policies the comprise the *Novice*, *Mixed* and *Experienced* multi-demonstrator datasets, including the return of the policy. Each individual demonstrator contributed 20,000 transition records. *Rand* denotes a random policy, *SL* a policy trained using the Softlearning library, and *D3* a policy trained using the D3RLPY library.

| Environment | Policy Identifier | Multi-Demonstrator Dataset | | | | | | | | |
| | | Novice | | | Mixed | | | Experienced | | |
| | | Type | Steps | Return | Type | Steps | Return | Type | Steps | Return |
|---|---|---|---|---|---|---|---|---|---|---|
| HalfCheetah | 1 | Rand | - | - | Rand | - | - | SL | 1 | 11032 |
| HalfCheetah | 2 | SL | 0.1 | 5126 | SL | 0.25 | 7663 | SL | 2 | **14511** |
| HalfCheetah | 3 | SL | 0.25 | **7663** | SL | 1 | **11032** | SL | 3 | 14215 |
| HalfCheetah | 4 | D3 | 0.1 | 3744 | D3 | 0.2 | 5050 | D3 | 1 | 8300 |
| HalfCheetah | 5 | D3 | 0.2 | 5050 | D3 | 2 | 9022 | D3 | 2 | 9022 |
| Hopper | 1 | Rand | - | - | Rand | - | - | SL | 0.6 | 3051 |
| Hopper | 2 | SL | 0.1 | 667 | SL | 0.2 | 3239 | SL | 0.8 | 3524 |
| Hopper | 3 | SL | 0.2 | **3239** | SL | 1 | **3553** | SL | 1 | **3553** |
| Hopper | 4 | D3 | 0.1 | 2567 | D3 | 0.2 | 1974 | D3 | 0.8 | 2625 |
| Hopper | 5 | D3 | 0.2 | 1974 | D3 | 1 | 3179 | D3 | 1 | 3179 |
| Walker2D | 1 | Rand | - | - | Rand | - | - | SL | 1 | 4139 |
| Walker2D | 2 | SL | 0.1 | 319 | SL | 0.25 | 1225 | SL | 2 | 4134 |
| Walker2D | 3 | SL | 0.25 | 1225 | SL | 1 | 4139 | SL | 3 | 4795 |
| Walker2D | 4 | D3 | 0.1 | 369 | D3 | 0.2 | 1587 | D3 | 1 | 4312 |
| Walker2D | 5 | D3 | 0.2 | **1587** | D3 | 2 | **5430** | D3 | 2 | **5430** |

## C  Environment Model OOD Performance

Figure 3 shows the average and worst-case log-likelihoods and mean-squared errors (MSEs) obtained for environment models trained on all MuJoCo environments and multi-demonstrator datasets. The performance metrics were calculated using datasets generated by demonstrators that did not contribute to the multi-demonstrator dataset used to train the model. Across all environments and multi-demonstrator datasets, environment models trained using REx achieved improved average and worst-case log-likelihoods. The results were more mixed for the MSE values–small increases in MSE were sometimes observed, as can be seen for the Walker2d *Experienced* dataset in Figure 3f. The discrepancy likely arises given it was the variance of log-likelihoods across training domains that was minimised during model training. Future experiments could evaluate the impact of updating the training procedure to minimise the variance of the training MSEs, or both the log-likelihoods and MSEs simultaneously.

Increasing the REx penalty coefficient leads to higher log-likelihood values in the majority of cases. One exception is the peak in worst-case performance that all Hopper multi-demonstrator datasets exhibit when $\beta = 10$. This indicates that the REx penalty coefficient is still a hyperparameter that needs to be appropriately tuned.

### C.1  Correlation between Environment Model OOD Performance and Policy Returns

As demonstrated by Figure 4, no correlation is observed between the average OOD log-likelihood of environment models and the average evaluation return of the policies trained using them. It should, however, be noted that the distribution of roll-out starting locations differs across the experiments (given that starting locations are drawn from the dataset used to train the environment model), which will have had an influence on the observed results.

## D  Policy Training Results

### D.1  ID Roll-out Starting Locations

The results of experiments run against multi-demonstrator datasets generated using the HalfCheetah, Hopper and Walker2d environments are shown in Table 4, Table 5 and Table 6 respectively. In all of these experiments, during policy training, roll-out starting locations were drawn from the same dataset used to train the environment model.

Across all environments, a policy trained offline with DIMORL obtained a higher average return for the *Novice* multi-demonstrator dataset than any of the individual demonstrators that were used to

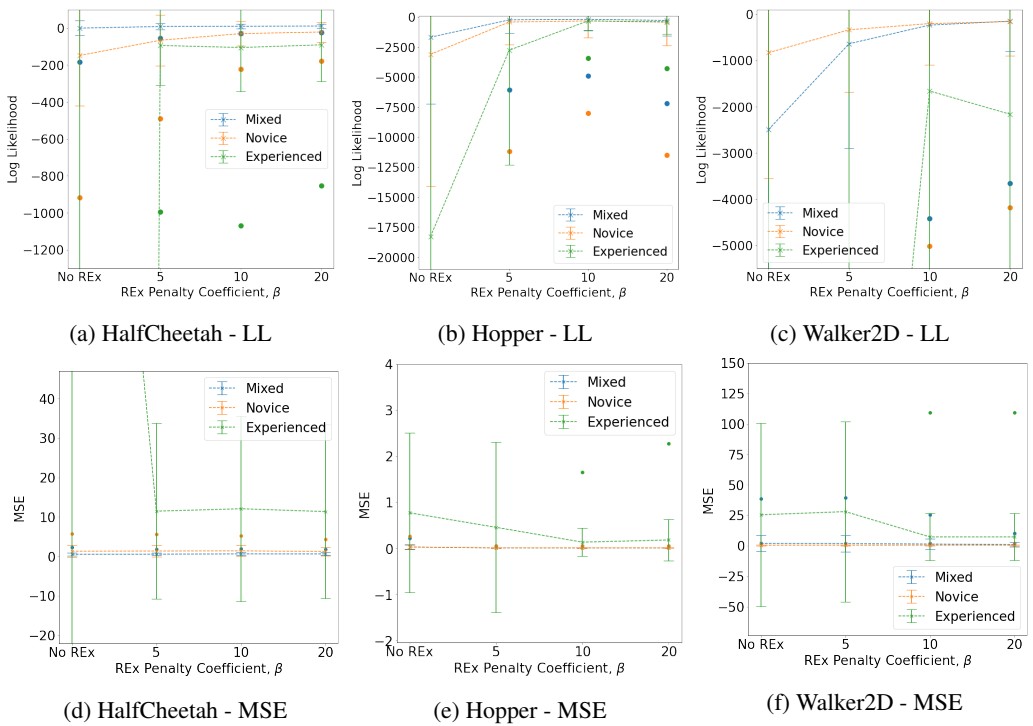

Figure 3: Environment models trained with REx typically achieve improved average and worst-case out-of-distribution performance, both in terms of log-likelihood (LL) and mean-squared error (MSE).

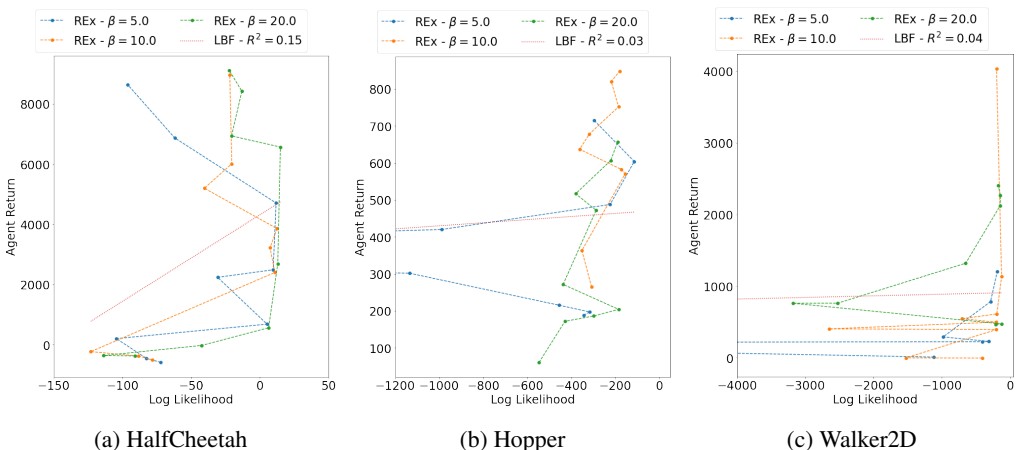

Figure 4: No correlation is observed between the average OOD log-likelihood of environment models and the average evaluation returns of policies trained using them. The results shown are for a roll-out length of 5. A line of best fit (LBF) is shown, along with the corresponding $R^2$ value.

generate the dataset. The same consistency was not observed for the *Mixed* or *Experienced* datasets. This reaffirms the importance of action diversity in offline model-based methods, whereas model-free techniques are potentially better suited to datasets collected from less-diverse, more experienced demonstrators, as was highlighted by Yu et al. [28].

We also note the lack of consistency in the results regarding the optimal MOPO penalty coefficient. Yu et al. [29] highlight the poor calibration between the MOPO penalty and actual model error, which is likely at least partly responsible for the inconsistency observed.

Table 4: Average and standard deviation, over three random seeds, of the evaluation returns for policies learned using dynamics models trained against multi-demonstrator datasets generated from the HalfCheetah environment. The overall highest return for each multi-demonstrator dataset has been bolded. If a value has a number in brackets next to it then only this number of agents completed 0.5 million steps of training. If there is no value then none of the agents completed 0.5 million steps of training.

| Dataset | Roll-out Length | MOPO Pen. Coeff, $\lambda$ | REx Penalty Coefficient, $\beta$ | | | |
|---|---|---|---|---|---|---|
| | | | 0 | 5 | 10 | 20 |
| Novice | 5 | 0 | 8591 ± 417 | 5917 ± 2700 | 6723 ± 1616 | 8157 ± 908 |
| Novice | 5 | 1 | **9056 ± 122** | 7188 ± 1800 | 7591 ± 647 | 6352 ± 3320 |
| Novice | 5 | 5 | 7134 ± 263 | 3066 ± 2962 | 2242 ± 3340 | 4349 ± 3135 |
| Novice | 10 | 0 | 5511 ± 4172 | 6553 ± 1776 | 2822 ± 2288 | 4235 ± 3419 |
| Novice | 10 | 1 | 8977 ± 190 | 6774 ± 2508 | 6705 ± 1704 | 5719 ± 4102 |
| Novice | 10 | 5 | 7004 ± 312 | 2259 ± 3344 | -197 ± 157 | 2188 ± 3283 |
| Novice | 20 | 0 | 4062 ± 4340 (2) | 6344 ± 2459 | 2772 ± 2399 | 5410 ± 2667 (2) |
| Novice | 20 | 1 | 4207 ± 4554 | 7392 ± 1953 | 5662 ± 1463 (2) | 4070 ± 4113 (2) |
| Novice | 20 | 5 | 3285 ± 3705 (2) | 2329 ± 3531 | -111 ± 39 (2) | 3078 ± 3147 (2) |
| Novice | 50 | 0 | -205 ± 0 (1) | 3700 ± 133 (2) | 4210 ± 902 (2) | 3839 ± 4340 (2) |
| Novice | 50 | 1 | 4219 ± 4410 (2) | 4847 ± 0 (1) | 5602 ± 221 (2) | 4249 ± 4182 (2) |
| Novice | 50 | 5 | 3537 ± 3802 (2) | -82 ± 0 (1) | -152 ± 17 | 2842 ± 3143 (2) |
| Mixed | 5 | 0 | 4961 ± 3372 | 2627 ± 1646 | 3160 ± 597 | 3273 ± 2486 |
| Mixed | 5 | 1 | **6687 ± 4888** | 4230 ± 1853 | 3195 ± 1720 | 4579 ± 77 |
| Mixed | 5 | 5 | 2374 ± 3472 | 6633 ± 499 | 4973 ± 3564 | 5161 ± 3728 |
| Mixed | 5 | 10 | -197 ± 73 | 2362 ± 2944 | 2054 ± 3012 | 477 ± 445 |
| Mixed | 5 | 20 | -131 ± 269 | 1588 ± 2415 | 84 ± 281 | 397 ± 136 |
| Mixed | 10 | 0 | -369 ± 93 | 1329 ± 1448 (2) | 1983 ± 1172 | 3087 ± 527 |
| Mixed | 10 | 1 | -211 ± 147 | 3906 ± 34 (2) | 3592 ± 805 | 3843 ± 966 |
| Mixed | 10 | 5 | -379 ± 360 | 4464 ± 3360 | 4616 ± 3467 | 4722 ± 3401 |
| Mixed | 10 | 10 | -67 ± 280 | 1637 ± 2564 | 2381 ± 3463 | 3904 ± 2862 |
| Mixed | 10 | 20 | -359 ± 219 | -196 ± 65 | 135 ± 250 | -557 ± 238 |
| Experienced | 5 | 0 | 40 ± 714 | -279 ± 344 | -369 ± 112 | -252 ± 161 |
| Experienced | 5 | 1 | 1607 ± 2549 | -348 ± 83 | -145 ± 125 | -331 ± 21 |
| Experienced | 5 | 5 | 1424 ± 1799 | 1453 ± 2458 | 3612 ± 979 | 3629 ± 747 |
| Experienced | 10 | 0 | -461 ± 17 (2) | -404 ± 97 | -455 ± 114 | -344 ± 96 |
| Experienced | 10 | 1 | -178 ± 48 | -665 ± 248 | -128 ± 125 | -301 ± 107 |
| Experienced | 10 | 5 | -239 ± 42 (2) | 1227 ± 2208 | **4547 ± 288** | 3492 ± 614 |
| Experienced | 20 | 5 | -56 ± 0 (1) | 634 ± 1594 | 3291 ± 2146 | 2958 ± 1644 |

## D.2 OOD Roll-out Starting Locations

We refer to roll-out starting locations as being OOD if they are sampled from any dataset other than the one used to train the environment model that is used in policy training. The results of experiments run against multi-demonstrator datasets generated using the HalfCheetah and Hopper environments are shown in Table 7 and Table 8 respectively. Roll-out starting locations for these experiments were drawn from the following datasets:

1. **RAND:** Transitions were generated by taking random actions in the environment.
2. **D4RL-MR:** The D4RL medium-replay datasets for each of the environments [9].

Experiments were not run for environment models trained on the HalfCheetah *Experienced* multi-demonstrator dataset given that (when using MOPO penalty coefficient $\lambda = 0$) no policies were learned when drawing starting locations from the same dataset used to train the models. If policies cannot be learned using in-distribution starting locations, then we do not expect to be able to learn them using OOD starting locations. Further, we are yet to run similar experiments for the Walker2D multi-demonstrator datasets, but we anticipate similar results to those seen for the HalfCheetah and Walker2D datasets (i.e., an inability to learn a policy). Additionally, experiments are still to be run for MOPO penalty coefficients $\lambda > 0$.

## E Degenerate Reward Predictions

Figure 5 demonstrates that abnormally large reward predictions can occur within the first 10 steps of episodes generated using environment models trained without REx. Out of the 60 episodes shown

Table 5: Average and standard deviation, over three random seeds, of the evaluation returns for policies learned using dynamics models trained against multi-demonstrator datasets generated from the Hopper environment. The overall highest return for each multi-demonstrator dataset has been bolded. If a value has a number in brackets next to it then only this number of agents completed 1 million steps of training. If there is no value then none of the agents completed 1 million steps of training.

| Dataset | Roll-out Length | MOPO Pen. Coeff, $\lambda$ | REx Penalty Coefficient, $\beta$ | | | |
|---|---|---|---|---|---|---|
| | | | 0 | 5 | 10 | 20 |
| Novice | 5 | 0 | 1184 ± 772 | 200 ± 11 | 483 ± 242 | 210 ± 44 |
| Novice | 5 | 1 | 1386 ± 1444 | 1240 ± 550 | 567 ± 154 | 1104 ± 416 |
| Novice | 5 | 5 | 502 ± 214 | 788 ± 90 | 1473 ± 1423 | 1305 ± 1560 |
| Novice | 10 | 0 | 978 ± 69 | 1558 ± 832 | 1323 ± 395 | 1686 ± 815 |
| Novice | 10 | 1 | 1796 ± 320 | 1506 ± 274 | 2313 ± 938 | 1829 ± 694 |
| Novice | 10 | 5 | 3117 ± 488 | 3073 ± 335 | 3056 ± 516 | 3471 ± 25 |
| Novice | 20 | 0 | 2591 ± 1100 | 2412 ± 840 | 2052 ± 992 | 2596 ± 600 |
| Novice | 20 | 1 | 2350 ± 1270 | 1804 ± 1165 | 2356 ± 223 | 3254 ± 303 |
| Novice | 20 | 5 | 3001 ± 622 | 2621 ± 1165 | 2788 ± 932 | 3449 ± 15 |
| Novice | 50 | 0 | 2437 ± 1081 | 2570 ± 1331 | 2761 ± 981 | 2145 ± 909 |
| Novice | 50 | 1 | 2467 ± 1165 | 2809 ± 724 | 2708 ± 1097 | 3461 ± 26 |
| Novice | 50 | 5 | 2527 ± 1236 | 2632 ± 845 | 2762 ± 1017 | **3482 ± 23** |
| Novice | 100 | 5 | 3411 ± 0 (1) | 3489 ± 12 (2) | 3220 ± 396 | 3444 ± 0 (1) |
| Mixed | 5 | 0 | 586 ± 128 | 602 ± 93 | 667 ± 128 | 459 ± 189 |
| Mixed | 5 | 1 | 664 ± 142 | 781 ± 257 | 574 ± 137 | 781 ± 24 |
| Mixed | 5 | 5 | 645 ± 119 | 749 ± 28 | 738 ± 34 | 1016 ± 150 |
| Mixed | 10 | 0 | 800 ± 33 | 876 ± 170 | 993 ± 325 | 1782 ± 1160 |
| Mixed | 10 | 1 | 1318 ± 622 | 911 ± 163 | 950 ± 468 | 919 ± 116 |
| Mixed | 10 | 5 | 1398 ± 639 | 1923 ± 1101 | 1491 ± 1481 | 2953 ± 518 |
| Mixed | 20 | 0 | 2521 ± 356 | 2706 ± 1232 | 2936 ± 823 | 1928 ± 731 |
| Mixed | 20 | 1 | 2208 ± 771 | 1988 ± 1149 | 1916 ± 780 | 2369 ± 857 |
| Mixed | 20 | 5 | 3483 ± 85 | 3558 ± 24 | 2864 ± 973 | 3320 ± 289 |
| Mixed | 50 | 0 | 2607 ± 863 | 2323 ± 1100 | 2010 ± 1146 | 2983 ± 724 |
| Mixed | 50 | 1 | 1443 ± 141 | 2940 ± 918 | 2598 ± 677 | 2767 ± 1121 |
| Mixed | 50 | 5 | 2839 ± 1049 | 3560 ± 48 | 3542 ± 50 | **3569 ± 40** |
| Mixed | 100 | 5 | 3226 ± 0 (1) | 3529 ± 33 | 3142 ± 609 | 3445 ± 156 |
| Experienced | 5 | 0 | 638 ± 231 | 355 ± 49 | 689 ± 48 | 380 ± 232 |
| Experienced | 5 | 1 | 607 ± 205 | 505 ± 254 | 316 ± 184 | 225 ± 107 |
| Experienced | 5 | 5 | 517 ± 200 | 304 ± 194 | 538 ± 104 | 601 ± 73 |
| Experienced | 10 | 0 | 864 ± 74 | 542 ± 204 | 757 ± 114 | 904 ± 113 |
| Experienced | 10 | 1 | 971 ± 393 | 429 ± 593 | 827 ± 45 | 842 ± 122 |
| Experienced | 10 | 5 | 1813 ± 1214 | 1191 ± 972 | 781 ± 85 | 683 ± 162 |
| Experienced | 20 | 0 | 2199 ± 811 | 1735 ± 1287 | 517 ± 354 | 1297 ± 239 |
| Experienced | 20 | 1 | 2472 ± 1024 | 868 ± 447 | 880 ± 130 | 984 ± 168 |
| Experienced | 20 | 5 | 1030 ± 640 | 1632 ± 1343 | 1006 ± 44 | 1774 ± 1171 |
| Experienced | 50 | 0 | 2811 ± 904 | 1424 ± 1528 | 960 ± 745 | 1756 ± 1035 |
| Experienced | 50 | 1 | 2479 ± 1002 | 2098 ± 1068 | 931 ± 663 | 1242 ± 134 |
| Experienced | 50 | 5 | **3493 ± 33** | 1343 ± 1142 | 3441 ± 52 | 2443 ± 782 |
| Experienced | 100 | 5 | 3318 ± 0 (1) | 334 ± 0 (1) | - | - |

Table 6: Average and standard deviation, over three random seeds, of the evaluation returns for policies learned using dynamics models trained against multi-demonstrator datasets generated from the Walker2d environment. The overall highest return for each multi-demonstrator dataset has been bolded. If a value has a number in brackets next to it then only this number of agents completed 3 million steps of training. If there is no value then none of the agents completed 3 million steps of training.

| Dataset | Roll-out Length | MOPO Pen. Coeff, $\lambda$ | REx Penalty Coefficient, $\beta$ | | | |
|---|---|---|---|---|---|---|
| | | | 0 | 5 | 10 | 20 |
| Novice | 5 | 0 | 263 ± 203 | 412 ± 263 | 501 ± 89 | 1068 ± 849 |
| Novice | 5 | 1 | 214 ± 222 | 98 ± 141 | -2 ± 1 | -2 ± 0 |
| Novice | 5 | 5 | -2 ± 0 | -3 ± 0 | -2 ± 0 | -3 ± 0 |
| Novice | 10 | 0 | 327 ± 62 | 521 ± 182 | 817 ± 242 | **1594 ± 1598** |
| Novice | 10 | 1 | 83 ± 121 | 325 ± 364 | 82 ± 118 | 86 ± 125 |
| Novice | 10 | 5 | -3 ± 0 | -2 ± 1 | -2 ± 0 | -3 ± 0 |
| Mixed | 5 | 0 | -1 ± 3 (2) | 500 ± 508 | 1721 ± 1700 | **2265 ± 141 (2)** |
| Mixed | 5 | 1 | 1 ± 6 | 1544 ± 2187 | 245 ± 349 | -2 ± 0 |
| Mixed | 5 | 5 | -3 ± 0 | -3 ± 0 | -3 ± 0 | -3 ± 0 |
| Mixed | 10 | 0 | 2050 ± 1922 | 1614 ± 2099 | 1095 ± 704 | 1653 ± 1563 |
| Mixed | 10 | 1 | -2 ± 0 | 1501 ± 2125 | 273 ± 390 | -2 ± 0 |
| Mixed | 10 | 5 | -3 ± 0 | -3 ± 0 | -2 ± 1 | -2 ± 1 |
| Experienced | 5 | 0 | 1685 ± 1850 | 144 ± 10 | 316 ± 232 | 947 ± 263 |
| Experienced | 5 | 1 | -3 ± 0 (2) | 2318 ± 2326 (2) | 1297 ± 1839 | 3080 ± 2202 |
| Experienced | 5 | 5 | -3 ± 0 | -4 ± 0 | -3 ± 0 | -4 ± 0 |
| Experienced | 10 | 0 | 3245 ± 1375 | 3121 ± 1667 | 3524 ± 1791 | **4796 ± 113** |
| Experienced | 10 | 1 | 1602 ± 2270 | 4785 ± 126 | 979 ± 1389 | 1501 ± 2127 |
| Experienced | 10 | 5 | -3 ± 1 | -3 ± 0 | -3 ± 0 | -3 ± 0 |

Table 7: Average and standard deviation, over three random seeds, of the evaluation returns for policies learned using dynamics models trained against multi-demonstrator datasets generated from the HalfCheetah environment. During offline policy training with the SAC algorithm, roll-out starting locations were sampled from the indicated dataset. If a value has a number in brackets next to it then only this number of agents completed 0.5 million steps of training. If there is no value then none of the agents completed 0.5 million steps of training.

| Env. Model Train. Dataset | Roll-out Start. Loc. Dataset | Roll-out Length | REx Penalty Coefficient, $\beta$ | | | |
|---|---|---|---|---|---|---|
| | | | 0 | 5 | 10 | 20 |
| Novice | RAND-1 | 5 | 374 ± 24 (2) | -54 ± 496 | -710 ± 613 | -164 ± 216 |
| Novice | RAND-1 | 10 | -156 ± 180 (2) | 153 ± 0 (1) | -311 ± 6 (2) | -345 ± 9 (2) |
| Novice | D4RL-HC-MR | 5 | 155 ± 392 | -438 ± 77 | -354 ± 152 | -518 ± 192 |
| Novice | D4RL-HC-MR | 10 | 442 ± 625 (2) | -411 ± 124 | -485 ± 223 | -786 ± 370 |
| Mixed | RAND-1 | 5 | -339 ± 41 | -453 ± 64 (2) | -395 ± 42 (2) | -392 ± 40 |
| Mixed | RAND-1 | 10 | -485 ± 0 (1) | -360 ± 0 (1) | -361 ± 9 (2) | -369 ± 10 |
| Mixed | D4RL-HC-MR | 5 | -459 ± 209 | -598 ± 211 | -641 ± 58 | -638 ± 94 |
| Mixed | D4RL-HC-MR | 10 | -767 ± 378 (2) | -494 ± 200 (2) | -483 ± 114 | -665 ± 160 |

in Figure 5, six contained reward predictions with an absolute value exceeding one million. If these episodes are excluded then the largest absolute reward prediction was 12.29. No episodes generated absolute reward predictions greater than 12.29 in the first 5 steps, while 2 episodes did within the first 10 steps. Note that our aim in this experiment was merely to show the possibility of abnormally large reward predictions in the early stages of episodes, not prove that the problem is common/widespread.

## F   Additional Experiments

### F.1   Noisy Roll-out Starting Locations

To further assess the potential benefits of environment models trained with REx, we analysed the impact of adding noise to the roll-out starting locations used during offline policy training. Note that the environment models were trained on the original multi-demonstrator datasets (which have no noise), and starting locations were still sampled from the same datasets used to train the environment models. The only change was that noise with standard deviation $\sigma \in \{0.00, 0.01, 0.05, 0.10\}$

Table 8: Average and standard deviation, over three random seeds, of the evaluation returns for policies learned using dynamics models trained against multi-demonstrator datasets generated from the Hopper environment. During offline policy training with the SAC algorithm, roll-out starting locations were sampled from the indicated dataset. If a value has a number in brackets next to it then only this number of agents completed 1 million steps of training. If there is no value then none of the agents completed 1 million steps of training.

| Env. Model Train. Dataset | Roll-out Start. Loc. Dataset | Roll-out Length | REx Penalty Coefficient, $\beta$ | | | |
|---|---|---|---|---|---|---|
| | | | 0 | 5 | 10 | 20 |
| Novice | RAND-1 | 5 | 6 ± 1 | 8 ± 2 | 9 ± 2 | 8 ± 2 |
| Novice | RAND-1 | 10 | 5 ± 1 | 9 ± 1 | 8 ± 2 | 9 ± 3 |
| Novice | D4RL-H-MR | 5 | 179 ± 126 | 253 ± 175 | 281 ± 200 | 236 ± 165 |
| Novice | D4RL-H-MR | 10 | 245 ± 172 | 385 ± 27 | 398 ± 35 | 297 ± 170 |
| Mixed | RAND-1 | 5 | 5 ± 1 | 7 ± 2 | 7 ± 2 | 6 ± 2 |
| Mixed | RAND-1 | 10 | 7 ± 1 | 9 ± 2 | 8 ± 2 | 31 ± 36 |
| Mixed | D4RL-H-MR | 5 | 270 ± 70 | 102 ± 130 | 99 ± 133 | 82 ± 104 |
| Mixed | D4RL-H-MR | 10 | 311 ± 66 | 310 ± 15 | 310 ± 36 | 307 ± 1 |
| Experienced | RAND-1 | 5 | 10 ± 4 | 8 ± 2 | 9 ± 2 | 9 ± 2 |
| Experienced | RAND-1 | 10 | 16 ± 12 | 9 ± 3 | 12 ± 4 | 13 ± 3 |
| Experienced | D4RL-H-MR | 5 | 177 ± 132 | 355 ± 467 | 186 ± 121 | 313 ± 29 |
| Experienced | D4RL-H-MR | 10 | 261 ± 172 | 181 ± 54 | 332 ± 81 | 281 ± 64 |

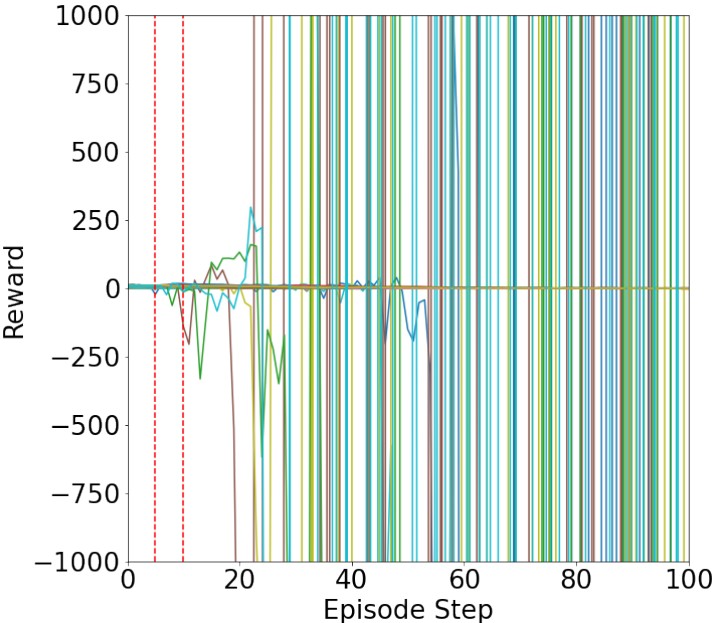

Figure 5: Abnormally large reward predictions can be made within the first 10 steps of an episode. Six different starting locations were randomly sampled from the HalfCheetah *Mixed* multi-demonstrator dataset. Ten episodes of 1000 steps were then generated from each starting location, using an environment model trained on the HalfCheetah *Mixed* multi-demonstrator dataset and a policy trained using the same environment model ($\beta = 0, \lambda = 0, h = 10$). For each episode, the rewards at each step over the first 100 steps are shown. The dashed vertical red lines denote the 5 and 10 step marks.

was added to the datasets before starting locations were sampled from them. Figure 6 shows that environment models trained with REx yielded policies with higher average returns across all noise levels for both the HalfCheetah and Hopper environments. For the Walker2D environment, environment models trained with REx penalty coefficients $\beta \in \{10, 20\}$ yielded policies with higher returns. For both HalfCheetah and Hopper environments, the highest policy evaluation return was obtained when $\sigma = 0.01$–suggesting that a small amount of noise was beneficial. We hypothesise that the distribution of noisy starting locations is more diverse, enabling greater exploration. In the case of the Hopper environment, it was the environment model trained without REx that obtained the highest average return for $\sigma = 0.01$, whereas the returns for the equivalent HalfCheetah and Walker2D environment models decreased significantly. This may be thanks to the greater simplicity of the Hopper environment.

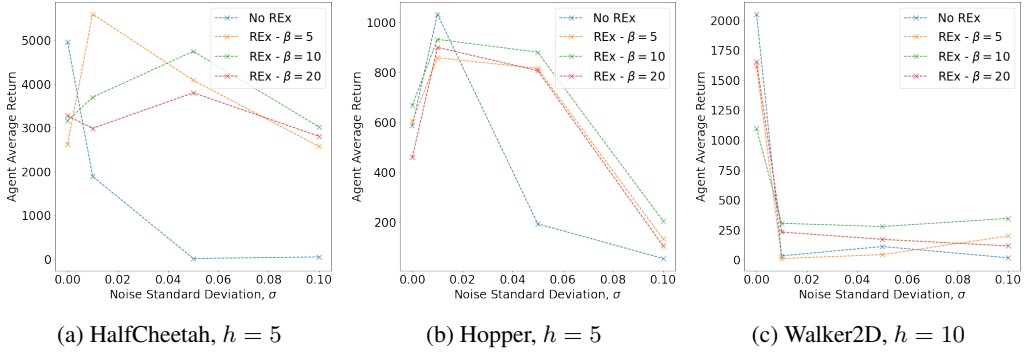

(a) HalfCheetah, $h = 5$        (b) Hopper, $h = 5$        (c) Walker2D, $h = 10$

Figure 6: The average returns of policies learned using environment models trained with REx display a greater robustness to noisy initial state distributions. Noise with standard deviation $\sigma \in \{0.00, 0.01, 0.05, 0.10\}$ was added to the roll-out starting locations used during offline policy training. Environment models were trained on the original multi-demonstrator datasets, which had no noise.

### F.2 Analysis of Training Exploration using PCA

In an attempt to determine whether environment models trained with REx supported increased exploration of the state-action space during policy training, we investigated the use of PCA to project transition records into two-dimensions, such that they could be visualised. For experiments using the Gym MuJoCo HalfCheetah environment, we sampled 100,000 transition records from the model pool every 100,000 training steps, excluding the zeroth and final training steps. Figure 7 shows the projections for a set of six policy training experiments, using the *Mixed* dataset and $\beta \in \{0, 10\}$. The projection matrix was trained on the complete collection of sampled transition records for the six experiments. Note that, unlike all other experiments presented in this paper, the environment models used in these experiments received one additional epoch of training prior to being used to learn a policy. This is the default behaviour of the MOPO codebase when utilising pre-trained environment models [28].

The average return for the experiments that did not use an environment model trained with REx was -420 ± 50, while for the REx experiments it was 8128 ± 1053. The projected records appear to indicate that increased exploration of the state-action space took place when using an environment model trained with REx. It is important to recognise, however, that the projections for REx experiments have higher explained variance, which may partially or fully explain the observed difference. The results will of course be sensitive to the datasets used to learn the projection matrix. We believe that further investigation of this method is warranted.

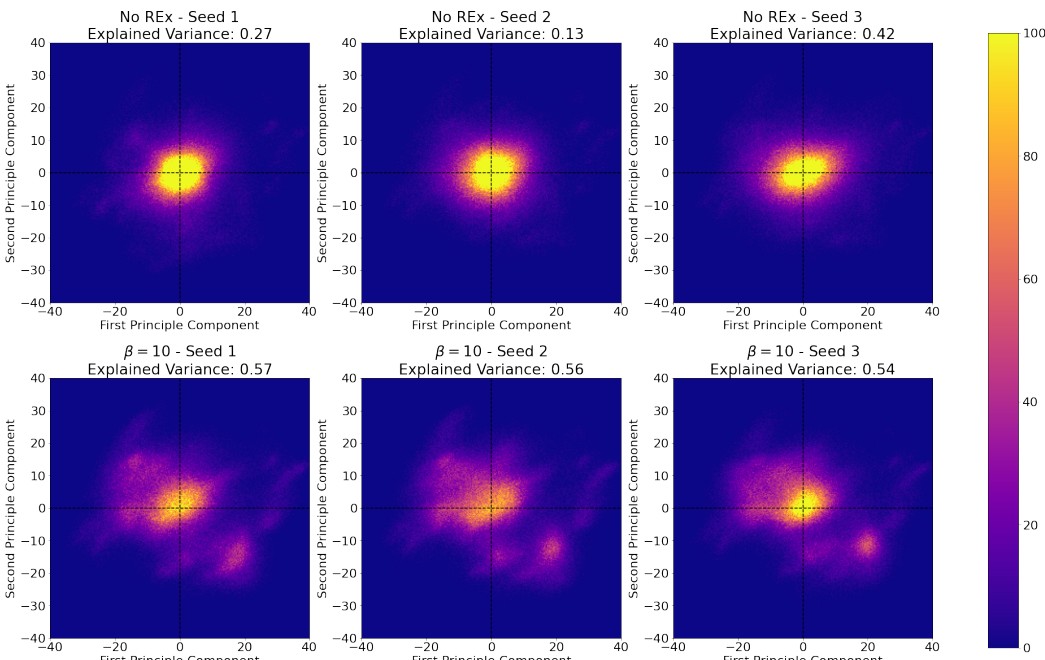

Figure 7: A 2D projection of state-action pairs sampled from the model pool during offline policy training appears to indicate that increased exploration of the state-action space took place when using an environment model trained with REx ($\beta = 10$). It is important to recognize, however, that the projections for the REx experiments have higher explained variance, which may partially or fully explain the observed differences. The projection matrix was trained on the complete collection of sampled transition records for the six experiments shown. MOPO penalty coefficient $\lambda = 5$ and roll-out length $h = 10$ were used.

