# OpenReview forum: "Domain Generalization for Robust Model-Based Offline Reinforcement Learning"
_NeurIPS.cc/2022/Workshop/Offline_RL — Offline RL Workshop NeurIPS 2022_

### Official Review · Reviewer_DdCd · 2022-10-19
**Interesting direction, can be improved by clearer motivation and stronger experiments**

**Rating:** 6
**Confidence:** 4

**Review:**

Summary
-------

This paper proposes to apply Risk Extrapolation (REx) to learn dynamics
and reward models from multiple demonstrators. This differs from the
usual application of REx, because the demonstrators are different
policies in the same MDP. The authors find that the learned models are
invariant to the particular policy providing the demonstration data,
allowing a policy to be trained on the learned model and attain better
acerage and worst case performance. The authors also note that the
REx-trained models increase stability.

Strengths
---------

-   Using REx in this problem is simultaneously simple and elegant. The
    differences between domains (different sources of data) and
    demonstrators is subtle, and there is potentially exciting work at
    this interface.

Weaknesses
----------

-   There is a lack of motivation in key design decisions in using REx
    over other domain generalization methods. A broader discussion of
    the alternatives, besides invariant risk minimization, is also
    missing.

-   The experiments should include stronger baselines, such as naive
    offline RL methods that do not do any domain generalization or other
    domain generalization methods. There are other unclear statements in
    the experiments, which I have detailed below.

Decision
--------

Overall, I think this paper studies an interesting problem. Some aspects
of the paper can be improved, and I encourage the authors to include
better motivation of the proposed domain generalization method for
multi-demonstrator RL, stronger baselines, and more rigorous
experiments.

Detailed Comments
-----------------

-   Section 3: The proposed method DIMORL assumes directly here that
    different demonstrators correspond closely to different domains.
    This is an interesting observation, and seems intuitively sound.
    There are, however, some important differences. For one, the
    transitions are all from a single MDP - which is the real domain,
    the state-action visitations provided by different policies provide
    a narrow slice of the global MDP. This additional structure is not
    exploited by REx, which is a missed opportuinity.

-   Figure 1: (b) I find this this figure very difficult to parse, are
    the intervals confidence intervals or max/min? For (a): What is the
    x-axis? And why isn't the same interval provided as in Figure (b)?

-   Section 3.1 (Rex Motivation): It is worth discussing some rationale
    for the selection of one domain generalization method over another.
    While you point to Invariant Risk Minimization paper as an
    alternative, no reasoning is provided for why it is not used which
    makes the motivation of the paper difficult to follow.

-   Section 3.1 (V-REx): It is not clear algorithm between V-REx and REx
    is used in the experiments.

-   Section 4 (pseudo- and multi- demonstrators): This type of
    demonstrator dataset provides a low diversity amongst policies for
    any particular strength. For example, the medium policies will
    likely all be very similar. It would be more interesting to have a
    collection of policies that are diverse even at a fixed level of
    strength (achieving similar return).

-   Table 1: The experiments can be strengthened by considering
    baselines that are either: (1) offline RL methods on the combined
    demonstrations (or indivudal per demonstration), and (2) other
    domain generalization methods. As it stands, there is no baseline,
    making the contribution difficult to evaluate.

-   Section 4.1 (log-likelihood results): It does not seem that the
    out-of-distriubtion log-likelihoods are reported anywhere in the
    main paper or the appendix. I am only able to find results reporting
    the policy return.

-   Section 4.2 (scale of beta): I would be curious to see how even
    larger values of beta perform because the best reported results are
    at the top end of the beta values considered. Does performance
    degrade gracefully as it is increased?

-   Section 4.2 (Clarification, OOD start-state): To clarify, the states
    sampled from the out-of-distribution datasets were completely random
    and not early on in a particular trajectory?

-   Section 4.3 (degenerate predictions): I do not follow how the
    evaluation returns in Figure 2b demonstrate the reward models can
    make dgenerate predictions, additional comments clarifying this
    statement should be in the text or the figure.

-   Section 4.3 (deadly triad and environment model): I do not think the
    deadly triad applies to this setting, because there is no
    bootstrapping in learning an environment model - it is an entirely
    supervised procedure.

---

### Official Review · Reviewer_XeF1 · 2022-10-19
**Good proof of concept for risk extrapolation applied to dynamics model learning**

**Rating:** 7
**Confidence:** 4

**Review:**

This paper considers the setting where offline RL is being performed from a multi-policy dataset, but it is known which policy collected which data. The authors argue that this poses something of a middle ground between two common settings: that when the dataset is generated by a single policy, and that when no assumptions are made and it is not known which policies generated which experiences.
The authors then connect this setting to domain generalization and propose using Risk Extrapolation applied to dynamics model learning in this setting. While preliminary, this work will be valuable to the workshop as a proof of concept.

My main suggestion to the authors would be to better contextualize the experiments with regard to common experimental settings. Though the experiments are described as an unusual setting for offline RL, from the appendix it appears that each of the datasets was collected from "subsets of five pseuo-demonstrators". My interpretation of this sentence was that each dataset consists of the experience of fewer than five policies. While this is technically different from the standard evaluation settings, the introduction made it seem like it would be more of a departure from, eg, the medium-expert D4RL datasets, and it was something of a disappointment to see that the datasets differed only minimally from those already in use.